# Transcriptome Alterations Caused by Social Defeat Stress of Various Durations in Mice and Its Relevance to Depression and Posttraumatic Stress Disorder in Humans: A Meta-Analysis

**DOI:** 10.3390/ijms232213792

**Published:** 2022-11-09

**Authors:** Vasiliy V. Reshetnikov, Polina E. Kisaretova, Natalia P. Bondar

**Affiliations:** 1Institute of Cytology and Genetics, Siberian Branch of Russian Academy of Sciences (SB RAS), Prospekt Akad. Lavrentyeva 10, Novosibirsk 630090, Russia; 2Department of Biotechnology, Sirius University of Science and Technology, 1 Olympic Avenue, Sochi 354340, Russia; 3Department of Natural Sciences, Novosibirsk State University, Pirogova Street 2, Novosibirsk 630090, Russia

**Keywords:** social defeat stress, depression, PTSD, prefrontal cortex, RNA-seq, gene expression

## Abstract

The research on molecular causes of stress-associated psychopathologies is becoming highly important because the number of people with depression, generalized anxiety disorder and posttraumatic stress disorders (PTSDs) is steadily increasing every year. Investigation of molecular mechanisms in animal models opens up broad prospects for researchers, but relevant molecular signatures can differ significantly between patients and animal models. In our work, we for the first time carried out a meta-analysis of transcriptome changes in the prefrontal cortex of C57BL/6 mice after 10 and 30 days of social defeat stress (SDS). We then examined possible correlations of these alterations with transcriptome changes found in post-mortem samples from patients with depression or PTSD. Although transcriptional signatures of human psychiatric disorders and SDS did not overlap substantially, our results allowed us to identify the most reproducible changes seen after SDS of various durations. In addition, we were able to identify the genes involved in susceptibility to SDS after 10 days of stress. Taken together, these data help us to elucidate the molecular changes induced by SDS depending on its duration as well as their relevance to the alterations found in depression or PTSD in humans.

## 1. Introduction

Depression is a common mental disorder; ~5% of the adult population (approximately 280 million people) is affected by this disease (Global Health Data Exchange—2019). Therefore, the study of molecular mechanisms of depression is an important and relevant task in modern neurobiology. It is known that stress increases the risk of depression [1,2], but the mechanism underlying this relation is not fully understood. According to social signal transduction theory of depression, experiences of social threat and adversity upregulate components of the immune system involved in inflammation, thereby leading to an increasingly proinflammatory phenotype, which may be a key phenomenon driving depression [3].

The use of animal models of depression opens up new opportunities for research on molecular changes in the brain. Chronic social defeat stress (chronic SDS)—an ethologically valid animal model of depression—induces a depression-like state in mice that is similar to depressive states in humans, with similarities in brain neurochemical changes and in symptoms, etiology, and sensitivity to antidepressants [4]. Molecular alterations in the brain after SDS substantially overlap with those observed in postmortem samples of various brain regions from patients with major depressive disorder (MDD) [5]. Additionally, as we showed previously [6], duration of SDS correlates with the severity of the depression-like state, in agreement with studies on people where cumulative stress throughout the life span is associated with a higher depression risk [7]. Prolonged exposure to SDS (21–30 days) leads to obvious hallmarks of depression, including a higher level of social avoidance, increased immobility in the forced swimming test, and anhedonic behavior as compared to the animals after 10 days of SDS [6,8]. On the other hand, after 10 days of SDS, only some animals (“susceptible rodents”) show behavioral changes, which manifest themselves as social withdrawal and increased anxiety; the other animals show only minor behavioral changes and represent the group of “resilient rodents” [9,10]. Of note, SDS can be regarded not only as a model of depression but also as a model of posttraumatic stress disorder (PTSD) because the stressed animals, in addition to the behavioral changes, exhibit intrusive symptoms (fear generalization) and hyperarousal symptoms (sleep fragmentation), which persist for a long time after exposure to stress and are characteristic of a model of PTSD [11].

The prefrontal cortex (PFC) is one of the most important brain structures involved in the regulation of depression and anxiety, as confirmed by numerous studies based on high-precision neuroimaging and optogenetic methods [12]. Many noninvasive methods (involving electrical and magnetic stimulation) and invasive methods (deep brain stimulation) that lead to PFC activation have an antidepressant effect even in treatment-resistant depression [13,14,15]. Projections from the ventral hippocampus, ventral tegmental area, and basolateral and basomedial amygdala to the PFC are implicated in anxiety and social avoidance [16,17,18,19]. PFC projections to the lateral habenula, dorsal raphe nucleus, and nucleus accumbens are involved in the regulation of social behavior, emotions, and stress adaptation [20,21]. Taken together, these data allow us to consider the PFC a crucial hub for the regulation of susceptibility/resilience toward stress and toward the development of depression.

Recent advances in next-generation sequencing technologies have made it possible to evaluate changes in gene expression levels across the genome. Despite the identification of specific changes in the expression of tens to hundreds of genes in different brain regions by next-generation sequencing, the molecular mechanisms of SDS have still not been fully elucidated. Differentially expressed genes (DEGs) vary greatly among many studies in this field, and this problem is caused by both methodological differences (e.g., in the duration of stress, the type of control group, time after the last procedure of SDS, and coordinates of the studied area in the brain) and differences in statistical analysis (method of data analysis and the confidence threshold).

Here, we performed the first meta-analysis of transcriptomic changes in the PFC after SDS of various durations. This analysis enabled us to identify clusters of the most reproducible changes in gene expression after 10 and 30 days of SDS and to determine which genes are associated with susceptibility and resistance to SDS after 10 days (Figure 1). Furthermore, we compared such changes in mice after SDS of various durations with data from a meta-analysis of postmortem transcriptomes of the dorsolateral PFC from depressed patients [22] and data from the first transcriptomic study on the dorsolateral PFC from patients with PTSD [23].

## 2. Methods

### 2.1. The Search for Studies According to PRISMA Guidelines

To find the most reproducible alterations in the transcriptome (RNA-seq) in the PFC after chronic SDS, we carried out a literature search on Google Scholar and PubMed (NCBI) according to PRISMA guidelines [24]. We used the following search query: (social defeat stress) AND (prefrontal cortex) AND (RNA-seq). This search was completed on 10 January 2022 and produced 407 hits on Google Scholar and 179 hits on PubMed. After screening of the papers, eligible studies were found to meet the following inclusion criteria: (1) the experiment was performed on mice of the C57BL/6 strain; (2) the model of repeated SDS (10–30 days) was used; (3) tissue samples of the cortex and/or PFC were examined; (4) RNA-seq as used for the gene expression analysis; and (5) raw data are available (Table 1). Exclusion criteria were: (1) non-original datasets (e.g., duplicate studies, re-analyses of pre-existing datasets); (2) non full-length article (conference paper); (2) not C57BL/6 strain; (3) RNA-seq analysis was not used; (4) prefrontal cortex tissue was not used; (5) SDS paradigm was not used; (6) the duration of the SDS was not 10 or 30 days; (8) no access to raw RNA-seq data. Two independent researchers (RVV and KEP) performed initial screening and quality assessment of the included studies. In total, we found eight studies that involved 10-day SDS and three studies where the stress model of 6, 15, and 30 days was examined (Figure 1). Three out of the eight studies where the 10-day stress model was used did not contain a link to raw data, and the data were not provided after a request to corresponding authors. Given that the duration of SDS can directly affect the observed molecular signatures [6], we decided to focus on the effects of stress after 10 and 30 days of SDS. In addition, our data (that were fully eligible to meet the criteria) on animals after 30 days of SDS (PRJNA846517) were included in the meta-analysis. All raw data included in our study were quality controlled. 

### 2.2. Reanalysis of the Published RNA-seq Data

We analyzed the effects of stress after 10 days of SDS regardless of susceptibility and resilience (five studies, 23 control tissue samples and tissue samples from 33 stressed animals) and after 30 days of SDS (two studies, eight tissue samples in the control group and 12 tissue samples in the stress group). Because in two studies, after 10 days of SDS, the stressed group was divided into “susceptible” and “resilient” animals, the most reproducible changes associated with these behavioral manifestations were evaluated too (nine tissue samples from “susceptible” animals and nine from “resilient” animals). Datasets for samples from different studies were combined and analyzed via a single protocol.

The sequencing data were preprocessed with fastp v0.20.1 [29] to remove adapters and low-quality sequences. The preprocessed data were mapped to the Mus musculus GRCm38 reference genome assembly by the HISAT2 aligner software, v2.2.1 [30]. HISAT2 alignments were quantified by means of featureCounts v2.0 [31]. 

The quality of the sequencing data was assessed using FastQC and Picard Collec-tRnaSeqMetrics software (Appendix A). The aligned data with fragments per kilobase of transcript per million fragments mapped (FPKM) > 0.1 were then converted into per-gene count tables by means of GENCODE vM22 gene annotation data. Genes were then subjected to an analysis of differential gene expression via the DESeq2 R-package [32]. Genes with an adjusted *p*-value (p-adj) < 0.05 were designated as statistically significant DEGs. 

For heterogeneity analysis, each study was put through differential expression analysis separately (DESeq2). Log2 fold change values and log2 fold change standard errors estimated by DESeq2 for each gene in each study were extracted and used as input for rma.uni function from metafor R package [33]. Cochrane’s Q test *p*-values (QEp) and I2 statistics were extracted from function output. A total of 3.87% of genes had QEp < 0.01 and inconsistency statistic I2 did not exceed 6% for any gene; based on these values we conclude that heterogeneity is not significant and using a fixed effect model (such as that used by DESeq2) is possible for estimating differential gene expression. In addition, we estimated coefficient of variation (CV) for each gene in each study separately and after unification using DESeq2 normalized counts (Appendix A).

### 2.3. Gene Set Enrichment Analysis (GSEA)

We performed GSEA to test whether the same Gene Ontology (GO) terms are enriched in DEG sets, and the same was done to down- or upregulated genes across studies. GSEA was conducted using the gseGO function of the ClusterProfiler (v4.0.5) R package. Genes were ranked by log2(Fold Change) from DESeq2 results. In the results, the normalized enrichment score indicated whether the genes were mostly up- or downregulated in a given gene set.

### 2.4. Functional Annotation of the Most Reproducible Genes That Are Associated with SDS

GO enrichment analysis was conducted using the enrichGO function from the Cluster-Profiler (v4.0.5) R package. Our dataset was tested for enrichment with genes specific to neurons, astrocytes, microglia, endothelial cells, oligodendrocytes, and oligodendrocyte precursor cells according to recently published data [34].

### 2.5. A Comparison of SDS-Related Genes and Genes Associated with Human MDD and PTSD

We compared the results of our meta-analysis regarding 10-day SDS and 30-day SDS with results of a meta-analysis of RNA-seq data obtained from three independent dorsolateral PFC (BA 8/9) datasets (postmortem tissue samples) of 79 MDD patients and 75 controls without MDD (Figure 2) [22]. Raw data of the studies included in the meta-analysis are available under GEO dataset IDs GSE102556, GSE101521, and GSE80655 [35,36,37]. For the comparison, a combined list was used that included 18 genes (p.adj < 0.1) associated with depression. Moreover, we compared SDS-related DEGs with DEGs of postmortem samples of dorsolateral PFC (BA 9/46) from PTSD patients (52 PTSD patients and 46 controls without MDD) [23]. Gene orthologs were identified by means of BioMart (https://www.ensembl.org/biomart/martview/, accessed on 1 June 2022).

## 3. Results

In total, the results of next-generation sequencing of 78 libraries were analyzed. The libraries on average contained ~48.6 million reads (range: 33.7–108.2 million), and all but two libraries contained less than 1% of ribosomal reads (Appendix A). Thus, all the libraries subjected to the meta-analysis had satisfactory quality. Aside from the meta-analysis, we analyzed each dataset separately (Appendix A). Our results indicated that between 23% and 100% of the DEGs (*p* < 0.05) that we identified were also found in the original studies (Appendix A). To evaluate the homogeneity of studies included in meta-analysis, principal component analysis (PCA) was performed. The results of PCA of the 1000 most variable or all expressed genes suggested that there were no strong outliers in the datasets (Figure 2).

### 3.1. Transcriptome Alterations Caused by 10-Day SDS

The meta-analysis of the five datasets from independent experiments involving 10-day SDS yielded 26 DEGs (15 upregulated and 11 downregulated, p.adj < 0.05, Appendix A). We noted that 10 days of SDS leads to upregulation of genes associated with neuroinflammation (*Vwf*, *Il1r1*, and *Il6ra*), hemoglobin genes (*Hbb-bt* and *Hbb-bs*), and melanocortin receptor 4 gene (*Mc4r*), which takes part in energy homeostasis. The expression of *Nes*, which codes for Nestin (a marker of neural stem cells and reactive astrocytes) was low. Expression of glucocorticoid-responsive genes changed in different directions (*Fkbp5* is upregulated, and *Hsd11b1* is downregulated). Analysis of the set of DEGs showed enrichment with seven GO terms (Appendix A), among which were “oxygene binding” (*Hbb-bt* and *Hbb-bs*) and “cytokine receptor activity” (*Il1r1* and *Il6ra*) (Figure 3). 

To determine the functional pathways or signatures of all expressed genes, we performed GSEA and found that SDS leads to expression activation of genes associated with GO terms (biological processes) “posttranscriptional regulation of gene expression,” “wound healing,” and “extracellular matrix organization” (Figure 3, Appendix A). Next, we tested whether the expression of 26 DEGs is more specific for individual cell types. The set of the DEGs was found to be enriched with endothelium-specific genes *Acer2*, *Il1r1*, *Vwf*, *Nes*, *Ctla2a*, and *Gypc* (p(χ^2^) = 0.001), possibly indicating an important role of endothelial cells in the response to SDS of moderate duration. Collectively, our data suggested that the most stable changes of gene expression induced by 10-day SDS are associated with glucocorticoid signaling, neuroinflammation, and oxidative stress.

### 3.2. Genes Responsible for Susceptibility to SDS

Our comparison of animals susceptible and resistant to SDS on the basis of two datasets revealed 102 DEGs (30 upregulated and 72 downregulated genes in susceptible mice compared to resilient ones). The majority (62%) of the downregulated genes are oligodendrocyte-specific (*p* < 0.001), and this set of DEGs is enriched with GO terms (biological processes) associated with myelination and ensheathment of neurons. In the susceptible mice, there was downregulation of key genes encoding myelin sheath proteins (Mbp, Mal, Mobp, and Plp1), a biosynthetic enzyme (Ugt8a), and a transcription factor (Olig2) involved in oligodendrocyte differentiation, and the same was true for other genes encoding important myelination proteins (Mag and Ermn). GSEA showed upregulation of genes related to GO terms “embryonic organ development” and “ensheathment of neurons.”

### 3.3. Transcriptome Alterations Caused by 30-Day SDS

As compared to 10-day SDS, the SDS for 30 days led to more pronounced changes in the PFC transcriptome. After an analysis of two cortical RNA-seq datasets from our research group, we found that 547 genes were differentially expressed (287 upregulated and 260 downregulated). Among the upregulated genes, the DEGs were predominant whose protein products are involved in transcription, translation, alternative splicing, and ATP synthesis. The key upregulated genes that are implicated in transcription processes are genes of RNA polymerases (*Polr2k* and *Polr2g*) and a histone-binding protein (*Rbbp4*); in translation processes are translation initiation factor genes (*Eif1* and *Eif2s2*); in processes of alternative splicing are genes of nuclear ribonucleoprotein (*Snrpa1*, *Snrpd1*, *Snrpd2*, *Snrpe*, and *Snrpg*); and in processes of ATP synthesis are genes of ATP synthases (*Atp5g3, Atp5e*, *Atp5j*, *Atp5k*, *Atp5l*, and *Atp5o*) and NADH dehydrogenses (*Ndufa1*, *Ndufb3*, *Ndufb4*, *Ndufc1*, *Ndufb9*, and *Ndufs6*). GO enrichment analysis and GSEA also highlighted pathways associated with the activation of transcription and translation. Overall, although these processes are energy-consuming, their activation is apparently required for cellular level adaptation to the conditions of prolonged chronic stress. Concurrent with the expression of transcription- and translation-related genes was upregulation of complement genes and microglial-reactivity genes including *Ifngr1*, *Il10rb*, *Cfcc*, *C1qb*, *Arhgd*, *Tyrobp*, *Aif1*, and *Mrc1*. The downregulated genes were mostly associated with NMDA receptor signaling (*Grin1*, *Grin2c*, and *Camk2g*), neuronal trafficking (*Htt* and *Disc1)*, and steroid receptor modulators (*Ncor2*, *Ncoa3*, and *Bcor*). 

### 3.4. The Comparison of DEGs Detected after MDD or PTSD in Humans and after SDS in Mice

A comparison of DEGs after 10 or 30 days of SDS with DEGs in postmortem samples of the dorsolateral PFC collected from MDD patients (18 DEGs, p.adj < 0.1) or patients with PTSD (651 DEGs, p.adj < 0.1) revealed that there is only a small number of genes in the overlap between the two datasets (Figure 3). Most of the DEGs detected in depressed samples are immediate early genes (*Npas4*, *Egr1*, *Egr2*, *Fos*, *Fosb*, *Nr4a1*, and *Nr4a3*). These genes code for transcription factors that regulate the expression of a large number of targets. We did not find an overlap between the set of DEGs of depressed humans and the set of DEGs of 10-day SDS. The same was true between the DEG set of 10-day SDS and the DEG set of 30-day SDS. Only the expression of *Nr4a3* was low according to both the meta-analysis of depressed patients and the meta-analysis of mice after 30-day SDS.

PTSD was found to cause a change in the expression of more genes (as compared to MDD), and, for this reason, the number of DEGs in the overlap with SDS DEG sets was also greater. Three genes (*Il1r1*, *Fkbp5*, and *Depp1*) were upregulated both after 10 days of SDS and in the postmortem PFC samples from patients with PTSD. A comparison of DEGs detected after PTSD with DEGs detected after 30 days of SDS uncovered 27 genes common between these datasets; among these genes, the expression of 22 genes changed in the same direction when the two datasets were compared. Collectively, our results showed that despite the presence of overlaps between the DEG sets of SDS and postmortem DEG sets detected after MDD or after PTSD, the gene networks involved in these phenomena are different (Figure 4).

## 4. Discussion

Our meta-analysis of publicly available RNA-seq data revealed that SDS of various durations causes changes in the expression of diverse genes in the PFC of mice. These alterations overlap only slightly with changes in gene expression seen in postmortem samples of the dorsolateral PFC from patients with MDD or patients with PTSD.

Ten-day SDS altered the expression of only a small number of genes in the PFC. In all likelihood, this is due to differences among the studies included in the meta-analysis. Although we used only RNA-seq data from 10-day SDS in C57BL/6 mice, some experiments differ in the methods of dissection of the PFC, the duration of confrontation with an aggressor, the time point of euthanasia after the last confrontation and mice substrains. Thus, the expression changes that we found—in glucocorticoid-responsive genes (*Fkpb5* and *Hsd11b1*) and genes associated with neuroinflammation (*Vwf*, *Il1r1* and *Il6ra*) and oxidative stress (*Hbb-bt* and *Hbb-bs*)—can be considered the most reproducible: these alterations are almost unaffected by the variations of experimental design. Upregulation of hemoglobin genes is observed in many brain structures in response to chronic SDS [25,26,27,28,38,39] and appears to provide neuroprotection in response to oxidative stress [40]. An increase in *Fpkb5* expression and a decrease in *Hsd11b1* expression are aimed at suppressing chronically elevated glucocorticoid levels. FKBP5 is a cochaperone that restricts glucocorticoid receptor function by delaying nuclear translocation and by decreasing glucocorticoid receptor-dependent transcriptional activity [41]. 11β-HSD1 performs intracellular conversion of inactive glucocorticoid (11-dehydrocorticosterone) into active glucocorticoid corticosterone [42]. Glucocorticoids regulate the expression of many genes, induce inactivation of noncore activities, restrain inflammation, restrict growth, and improve the efficiency of energy production [43]. Nonetheless, chronically increased basal glucocorticoid levels in rodents are associated with greater accumulation of proinflammatory markers in the CNS [44]. Deficiency in 11β-HSD1 attenuates the brain cytokine response to inflammation [45]. Overexpression of *Il6ra* and *Il1r* found after 10 days of SDS is also seen during inflammatory reactions and is associated with increased IL-1/IL-6 signaling and neuroinflammation [46,47]. Furthermore, *Il6r* is a glucocorticoid-sensitive gene; this observation reflects the close relationship between the processes of stress response and neuroinflammation [43,48]. Finally, SDS also led to upregulation of an endothelial marker (*Vwf*) that is associated with blood–brain barrier permeability and downregulation of nestin (*Nes*), a marker of reactive astrocytes, which participates in vascular repair and remodeling [49,50,51]. Altogether, these findings indicate that the most reproducible changes in gene expression after 10-day SDS are directed toward homeostasis, and various cell types are involved in these processes. To identify more specific molecular mechanisms of stress, many experiments will be needed that are based on a unified methodology of SDS. 

Of note, RNA-seq data on male DBA/2NCrl mice after 10 days of SDS [27] showed no DE genes (p.adj < 0.1). Data on female C57BL/6 mice [28] after 10 days of SDS showed only four DE genes (*Gm10284*, *Gm7027*, *Ciart*, *Snord87*; FPKM < 0.1, p.adj < 0.1); however, these genes do not match the most reproducible genes found by us. Thus, these data support the idea that molecular signatures of SDS are sex- and strain-specific.

Thirty-day SDS resulted in changes in the expression of a broader range of DEGs. Primarily, this is because the meta-analysis includes two studies of very similar design. Most of the upregulated DEGs are associated with protein biosynthesis. Activation of protein biosynthesis is an energy-consuming process [52] and can be switched on by the cell as a defense mechanism in response to oxidative stress in order to prevent cell death [53,54]. This notion is in good agreement with the overexpression of microglial-activation-related and neuroinflammation-related genes that we found, because these two processes are known to lead to oxidative stress [55]. Excessive activation of protein biosynthesis causes synaptic and behavioral aberrations, which are characteristic, for example, of autism [56]. 

Among the downregulated DEGs, there are genes of NMDA receptor subunits whose dysfunction is associated with depression [57]. The expression of genes *Htt* and *Disc1* was also low, and their downregulation correlates with the development of various neurodegenerative and neuropsychiatric diseases [58,59]. Protein products of these genes participate in the regulation of synaptic function, axon and dendritic transport (trafficking), and interactions of Disc1/Huntingtin-mediated BDNF transport in the cortico-striatal circuit [60]. Both disturbances of the BDNF–TrkB pathway and abnormalities in cortico-striatal circuits are characteristic of depression [61]. 

Next, we analyzed genes associated with stress susceptibility after 10 days of SDS. Stress susceptibility was found to be strongly associated with myelination. Disturbances in the processes of myelination, development, and differentiation of oligodendrocytes are often observed in various stress-related pathologies in both animals and humans [62]. Unexpectedly, the set of DEGs associated with susceptibility to stress overlapped only slightly with the set of genes whose expression changed after 10 or 30 days of SDS. Consequently, myelination is most likely linked with strategies of adaptation to adverse experiences but does not affect the magnitude of changes in the expression of stress-induced genes.

Our most interesting findings are about *Fkbp5*, which was upregulated in susceptible animals after SDS as compared to resilient ones. Furthermore, an increase in *Fkbp5* expression was observed after 10 days of SDS and in postmortem PFC samples from patients with PTSD. On the contrary, after 30 days of SDS and in post-mortem PFC samples from patients with MDD, *Fkbp5* expression was unaltered. These results seem to reflect the role of FKBP5 in a certain stage of adaptation to stress and in the development of psychiatric disorders. Human and animal studies point to a strong correlation between *FKBP5* gene variants and environmental factors (in particular, experience of stress early in life) in psychiatric disorders, thereby implying the participation of epigenetic mechanisms in the regulation of FKBP5 expression [63,64]. 

Finally, to answer the question about possible similarities in molecular changes between the PFC in mice after SDS of various durations and human PFC samples from patients with depression or PTSD, we compared the DEGs found in the transcriptome data after depression or PTSD in humans and DEGs from our meta-analysis of SDS in mice. We found that the overlap between the sets of DEGs from murine SDS and the postmortem human sets of DEGs is rather small, probably indicating that the mechanisms underlying these pathologies are different. Nonetheless, some of the DEGs—that had the same direction of expression change both in the mouse model of SDS and human postmortem PFC samples—are genes known to be connected with the development of mental disorders: *Fkbp5* [63,64], *Il1r* [65], *Lpr8* [66], and *Txnip* [67].

Of note, the comparison of post-mortem human data with animal data has a number of limitations. First, post-mortem studies have high heterogeneity (genetic background, differences in age, and differences in the duration and severity of a disease), small sample sizes, and lack of ethnic diversity [68] because they are composed primarily of subjects with European or North American genetic backgrounds [69]. Second, it is still unclear which prefrontal regions can be considered equivalent between mice and humans [70]. Another assumption is that the analysis was conducted on a combination of male and female samples (see the legend in Figure 4). Although the aim of this study was not to evaluate sex-specific effects, extensive data from various animal models of stress, PTSD, and human depression demonstrate strong sex-specific effects on both behavior and molecular characteristics [71,72,73,74,75].

## 5. Conclusions

Developments in next-generation techniques have helped to considerably expand our knowledge about stress-related gene networks, thereby clarifying the pathogenesis of stress-related mental disorders. At the same time, high sensitivity of such approaches to methodological variations has given rise to heterogeneous datasets with many false positive and false negative results. One possible solution to this problem is to conduct meta-analyses. Accordingly, our meta-analysis of RNA-seq data from the model of SDS of various durations made it possible to identify the most valid changes in the PFC transcriptome that are characteristic of this type of stress. Additionally, a comparison of the SDS datasets with data obtained from post-mortem PFC samples from MDD patients or from PTSD patients led to the conclusion that PFC transcriptome signatures overlap between these datasets rather modestly. It is possible that such an analysis of other brain structures or a study on other molecular mechanisms, including those linked with epigenetic signatures, will help to draw comprehensive conclusions about the relevance of the SDS model to the research on molecular mechanisms of depression and PTSD.

## Figures and Tables

**Figure 1 ijms-23-13792-f001:**
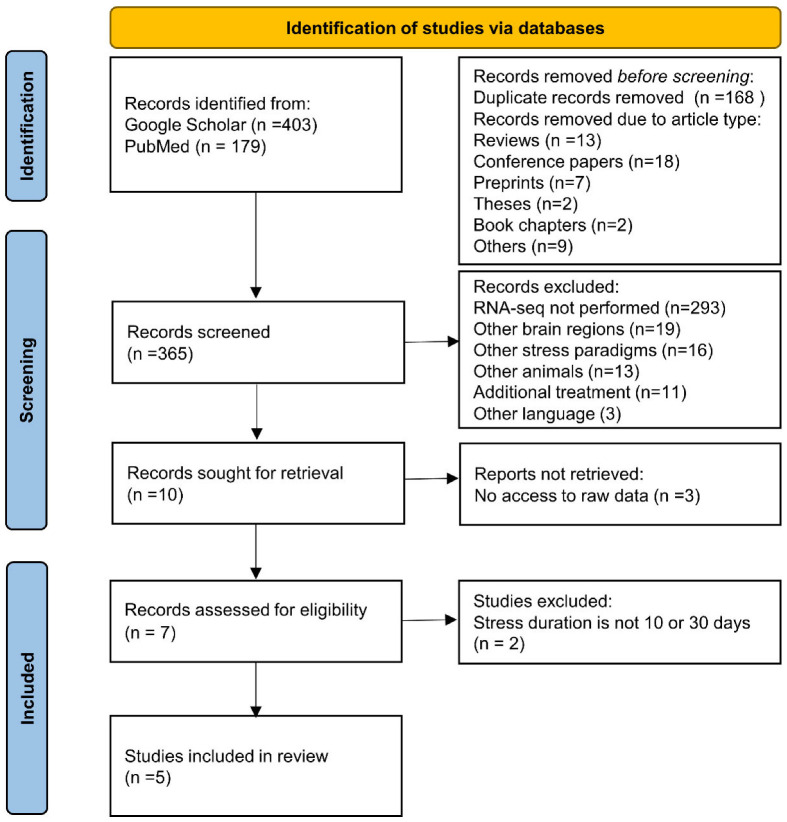
The PRISMA flow diagram.

**Figure 2 ijms-23-13792-f002:**
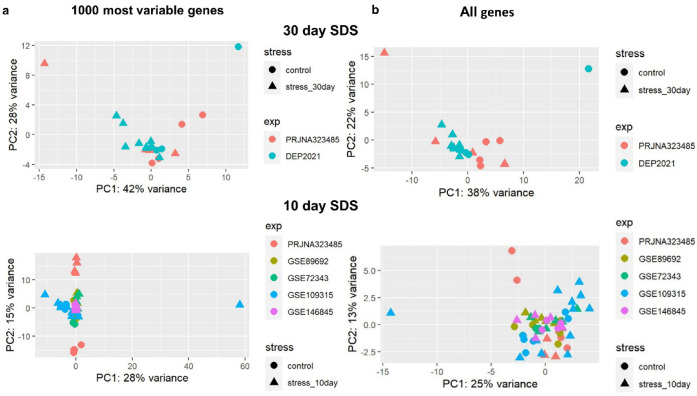
Principal component analysis (PCA) of all datasets included in meta-analysis. (**a**) Based on 1000 most variable genes. (**b**) Based on all expressed genes (FPKM > 0.1) The scatter plots were based on PCA scores of the first two principal components (PCs).

**Figure 3 ijms-23-13792-f003:**
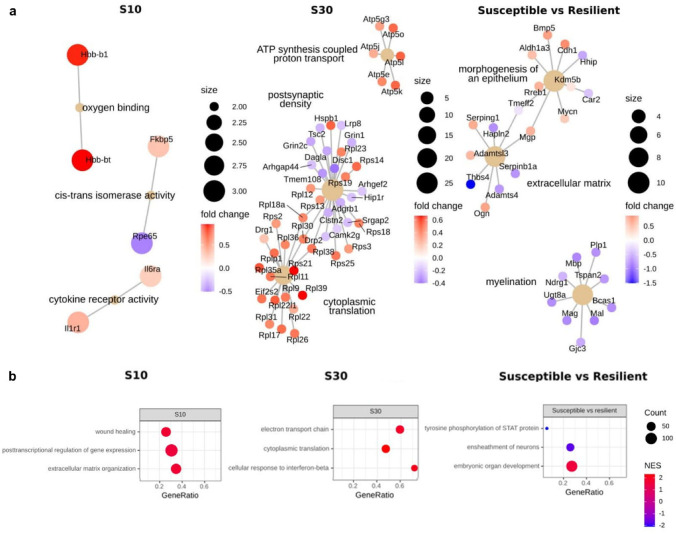
Functional annotation and GSEA of genes. (**a**) Top three most significant GO terms enriched in a set of DEGs (p.adj < 0.05) in different groups. (**b**) Top three most significant GSEA categories among all expressed genes (FPKM > 0.1) in different groups.

**Figure 4 ijms-23-13792-f004:**
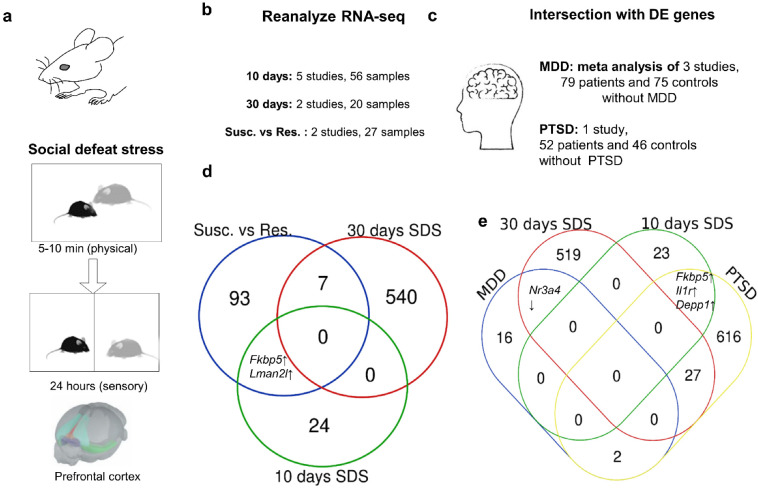
Experimental design. (**a**) Brief overview of the SDS paradigm. (**b**) Datasets of RNA-seq data on the PFC from SDS-exposed mice that were used for reanalysis. (**c**) Datasets of RNA-seq data on dorsolateral PFC (BA09) from postmortem samples from PTSD patients (52 PTSD: 26 males, 26 females; 46 control: 26 males, 20 females; age range > 18 to <70 years) and MDD patients (79 MDD: 49 males, 30 females; 75 controls: 57 males, 18 females; age range > 19 to <82 years) that were used for comparisons with SDS datasets. (**d**) A Venn diagram of DEGs (padj < 0.05) across different SDS datasets. (**e**) Comparisons of DEGs from the murine SDS data sets and from PTSD and MDD datasets of humans. Susc.: susceptible mice; Res.: resilient mice.

**Table 1 ijms-23-13792-t001:** The studies on SDS included in the meta-analysis.

Animal	Age	Duration of Chronic SDS	Type of Aggressor	Tissue Collection	Coordinates of PFC	Number of DEGs (*p* < 0.05)	Raw Data Project ID, Ref.	Data ID
♂ C57BL/6J	8–12 weeks	10 d	♂ CD1	PFC, 24 h after last defeat	Cortical area (0.75 to 3.25 mm from the bregma) was dissected out at an angle of approximately 30°	1273 ↑ 846 ↓	PRJNA323485 [6]	SRX1811104 SRX1809034 SRX1809033 SRX1809030 SRX1809039 SRX1809037 SRX1809036 SRX1809035
♂ C57BL/6J	6–8 weeks	10 d	♂ CD1	PFC, 4 days after last defeat	N/A	502 ↑ 443 ↓	GSE89692 [25]	GSM2386801 GSM2386802 GSM2386803 GSM2386804 GSM2386805 GSM2386806 GSM2386807 GSM2386808 GSM2386809 GSM2386810
♂ C57BL/6J	6–8 weeks	10 d	♂ CD1	PFC, 2 days after last defeat	N/A	1155 ↑ 1124 ↓	GSE72343 [26]	GSM3736027 GSM3736055 GSM3736063 GSM3736067 GSM3736073 GSM3736105 GSM3736029 GSM3736041 GSM3736069 GSM3736081 GSM3736107 GSM3736109
♂ C57BL/6NCrl	8 weeks	10 d	♂ CD1	mPFC, 5–8 days after last defeat	mPFC	556 ↑ 527 ↓	GSE109315 [27]	GSM2938660 GSM2938661 GSM2938662 GSM2938663 GSM2938664 GSM2938665 GSM2938666 GSM2938667 GSM2938668 GSM2938669 GSM2938670 GSM2938671 GSM2938672 GSM2938673 GSM2938674 GSM2938675 GSM2938676 GSM2938677
♂ C57BL/6J	10 weeks	10 d	♂ CD1	PFC, 5 days after last defeat	N/A	571 ↑ 511 ↓	GSE146845 [28]	GSM4407773 GSM4407774 GSM4407775 GSM4407776 GSM4407777 GSM4407782 GSM4407783 GSM4407784 GSM4407785
♂ C57BL/6J	8–12 weeks	30 d	♂ CD1	PFC, 24 h after last defeat	Cortical area (0.75 to 3.25 mm from the bregma) was dissected out at an angle of approximately 30°	1403 ↑ 1410 ↓	PRJNA323485 [6]	SRR3607976 SRR3607977 SRR3607964 SRR3607965 SRR3607963 SRR3611865 SRR3607966 SRR3607967
♂ C57BL/6J	8–12 weeks	30 d	♂ CD1	PFC, 24 h after last defeat	Cortical area (0.75 to 3.25 mm from the bregma) was dissected out at an angle of approximately 30°	1105 ↑ 1104 ↓	PRJNA846517	SRR19726009 SRR19726008 SRR19725997 SRR19725992 SRR19725987 SRR19725986 SRR19726007 SRR19726006 SRR19726005 SRR19726004 SRR19726003 SRR19726002

## Data Availability

The data that support the findings of this study are available from the corresponding author on request.

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
