# Peer review of "Transcriptome Alterations Caused by Social Defeat Stress of Various Durations in Mice and Its Relevance to Depression and Posttraumatic Stress Disorder in Humans: A Meta-Analysis"

_ijms, 2022, doi:10.3390/ijms232213792_

Round 1

Reviewer 1 Report (Previous Reviewer 2)

The resubmitted manuscript “Transcriptome alterations by social defeat stress of various durations in mice and its relevance to depression and post-traumatic stress disorder in humans: a meta-analysis” by Dr V. Reshetnikov and colleagues describes a comparison of transcriptional signatures in the PFC of social defeat stress (SDS) for 10 or 30 days in C57Bl/6 mice to those of the clinical human population with major depressive disorder (MDD) or post-traumatic stress disorder (PTSD). They found that longer exposure to SDS engendered greater transcriptional alterations, and that SDS-susceptible mice show disruption of myelin-related transcripts. Further, the authors show surprisingly little overlap between transcriptomic signatures from postmortem human PFC tissue from individuals with MDD or PTSD to the murine transcriptomic signatures following SDS. The studies described are well-designed, and the findings uncovered contribute to the field as a whole by directly comparing transcriptomic alterations following exposure to SDS in with the clinical population suffering from conditions thought to be modeled by SDS exposure. Further, the authors have included a number of [primarily statistical] edits to the resubmission that have improved the manuscript. However, some concerns from the initial submission remain and should be addressed.

Major concerns

The authors have not addressed studies included in the original submission that utilized either a different strain (Liane et al 2018) or biological sex (Deonaraine et al 2020) in the discussion, as strongly suggested following the initial submission.

The authors are encouraged to broaden the discussion of differences between murine studies to address sex and strain. In particular, biological sex should be discussed with regard to comparison to human data.

Importantly, the authors claim that Deonaraine et al 2020 examined male mice in Table 1, but this publication specifically looked at females. This should be corrected in the manuscript.

Minor concerns

- Indicate sex of mice for all studies in Table 1, as well as the sex of the aggressor mice.

- Include a table describing human subjects (age, sex, etc) within the 4 studies compared to murine data, or to add this information to Figure 3C.

Author Response

Major concerns

The authors have not addressed studies included in the original submission that utilized either a different strain (Liane et al 2018) or biological sex (Deonaraine et al 2020) in the discussion, as strongly suggested following the initial submission.

Reply: The comparison of our results with these data was noted in the Discussion section.

The authors are encouraged to broaden the discussion of differences between murine studies to address sex and strain. In particular, biological sex should be discussed with regard to comparison to human data.

Reply: You're absolutely right, the data from mixed samples of post-mortem male and female samples could be one of the reasons of why we found minor overlaps with data from animal models. We noted this as a limitation at the end of the Discussion section.

Importantly, the authors claim that Deonaraine et al 2020 examined male mice in Table 1, but this publication specifically looked at females. This should be corrected in the manuscript.

Reply: For our analysis, we only used data on males from the Deonaraine et al 2020 study. We checked if all animals used in our study were males and confirmed that this is true.

Minor concerns

- Indicate sex of mice for all studies in Table 1, as well as the sex of the aggressor mice.

Reply: We added aggressor gender symbols to the Table 1.

- Include a table describing human subjects (age, sex, etc) within the 4 studies compared to murine data, or to add this information to Figure 3C.

Reply: Thank you for the comment. We have added the necessary information to the figure caption in order not to clutter up the figure itself.

Reviewer 2 Report (New Reviewer)

In the review, Reshetnikov et al. carried out a meta-analysis of studies focusing on PFC transcription changes in C57BL/6 mice after 10 and 30 days of Social Defeat Stress (SDS), and compared those changes with post-mortem brains of patients that suffered from depression or post-traumatic stress disorder. The authors compared  openly available RNAseq raw data from peer-reviewed studies together with their own data. The authors found that the transcription changes between DEGs of SDS of 10 and 30 days and postmortem patients samples were different. The slight overlap between susceptible vs resilient mice and 10d SDS includes Fkbp5, a glucocorticoid receptor dependent cochaperone. Additionally, PTSD brain samples also showed increased Fkbp5 levels, which point to a correlation between glucocorticoid expression and stress in rat and clinical models. *Overall authors found…

This is an interesting and explanatory paper of genomic data that could further explain neurochemical correlations between different stress-associated psychopathologies, such as MDD and PTSD. The aim is retrospective and summatory as a meta-analysis and it is easy to follow and concise. We hope that the comments and issues I address help the authors move forward.

Major Issues

  1. PFC, unsure of what area of the cortex is being analyzed. We recommend the authors make mention of differences in location and the range of samples analyzed. 

  2. It is unclear if the sample used for SDS 30 has been peer reviewed. If not, all details must be provided to evaluate data on its own. 

Minor Issues

  1. Due to differences in protocols for SDS10 sample, we recommend making notice of differences in latency of euthanasia, methods of dissection of PFC, duration of confrontation with an aggressor and the point of euthanasia after the last confrontation. 

  1. In Figure 3a and in the Discussion, the authors do not make mention of the exposure and training that SDS implicates. Being that this is a meta-analysis and data from different papers is being analyzed, we would want to corroborate that the SDS protocol is the same for all papers being examined.

Author Response

PFC, unsure of what area of the cortex is being analyzed. We recommend the authors make mention of differences in location and the range of samples analyzed. 

Reply: Thanks for your comment. We have added the required information to Table 1. Unfortunately, some of the original studies did not clarify this information.

It is unclear if the sample used for SDS 30 has been peer reviewed. If not, all details must be provided to evaluate data on its own. 

 Reply: These data have not yet been published in a peer-reviewed article. The description of the project is presented at the corresponding link in NCBI (PRJNA846517). For raw data, quality control was performed on a par with others (data are presented in Supplementary Figure 1). The analysis of PRJNA846517 data was carried out according to the algorithm described in materials and methods.

Minor Issues

Due to differences in protocols for SDS10 sample, we recommend making notice of differences in latency of euthanasia, methods of dissection of PFC, duration of confrontation with an aggressor and the point of euthanasia after the last confrontation. 

 Reply: . We have added the required information to Table 1

In Figure 3a and in the Discussion, the authors do not make mention of the exposure and training that SDS implicates. Being that this is a meta-analysis and data from different papers is being analyzed, we would want to corroborate that the SDS protocol is the same for all papers being examined.

Reply:  Thanks for your comment. Yes, you're right, in some papers, various tests have indeed been applied to assess behavior. We make the assumption in the discussion section that they may have influenced outcome, but since the conditions of these tests were not aversive, their overall contribution should be negligible.

Reviewer 3 Report (New Reviewer)

The manuscript entitled “Transcriptome alterations caused by social defeat stress of various durations in mice and its relevance to depression and posttraumatic stress disorder in humans: a meta-analysis” by Reshetnikov and coworkers is focusing the stress-related gene networks that suppose to be involved in the pathogenesis of stress-related mental disorders. In the present form, the manuscript offers a variety of important information obtained in both animal experimental models and on human material.

Some potentially crucial alterations were observed for both differences induced by stress induction duration in preclinical trials that may be attributed to the stage of adaptation to stress. However, as the authors stated more accurate information could be achieved with the common experimental design that avoids the effect of experimental protocol variations.

Although the overlap with the results obtained on human material is not impressive, I still find it scientifically interesting, especially the role of FKBP5. 

The author should pay attention to technical preparation of the manuscript (different font size, etc.).

Author Response

Thank you for your review! We carefully searched for technical mistakes and made necessary corrections.

This manuscript is a resubmission of an earlier submission. The following is a list of the peer review reports and author responses from that submission.

Round 1

Reviewer 1 Report

The authors described a meta-analysis to determine shared differential gene expression (RNA-Seq) in prefrontal cortex between animal models of social defeat stress (SDS) with different timelines of stress (10 vs 30 d). The authors also compared the DEGs from SDS to RNA-Seq data from cortex of MDD & PTSD postmortem samples. The authors found very limited overlap both within individual datasets and between 10 and 30 d SDS. The authors attribute some of this to variations in the experimental differences (tissue collection, timeline after stress, etc.). The authors find virtually no overlap between the animal models and clinial MDD/PTSD postmortem samples. The authors point to potential differences that may explain the low degree of shared DEG, however the impact of these findings seem to be low. This paper would benefit from a higher amount of analysis of the findings, especially within datasets, that may explain the very low amount of overlap. There are concerns about the clinical postmortem samples that may have been used for this meta-analysis. Finally, because supplementary figures/tables were not available to this reviewer, this manuscript cannot be accurately peer-reviewed.

Major criticisms

The authors analyze RNA-Seq datasets from studies with widely ranging differences in experimental procedures (stress procedures, time of tissue collection after SDS, tissue collection methods, etc), which they do point out, however it would benefit to address the degree of overlap between datasets. In other words did multiple studies share the same DEGs, but these DEGs were excluded because one dataset did not? For example, within the 10 d SDS, did 4 of 5 studies have high overlap of DEGs, but the 5th did not therefore these DEGs were excluded. It is unclear in this met-analysis the degree of overlap between each dataset. Some analysis should be made to determine the relationship within a given set of datasets, e.g. 10 d SDS.

It is unclear why the authors chose only 3 clinical MDD datasets and only 1 clinical PTSD dataset. The authors describe a rigorous method (although see point below) to narrow down SDS datasets and how to integrate them. However, they make very little attempt to do the same with the clinical data. In addition, the MDD and PTSD gene expression data show almost no overlap, which is not consistent with the literature. The authors explain some of the limitations of the preclinical data but fail to describe the limitations of the postmortem data.

There is no included supplementary information.

Minor criticisms

The authors used a rigorous method to exclude findings, however some of the reasons for exclusion were not clear: ‘Records removed for other reasons’ during Identification and ‘Records Excluded’ during Screening. This should be explained.

Some minor grammatical errors:

·      Incorrect usage of hyphens throughout

·      I think besides is improperly used in this manuscript

·      Table 1 rows are not aligned

Author Response

Reviewer1

The authors described a meta-analysis to determine shared differential gene expression (RNA-Seq) in prefrontal cortex between animal models of social defeat stress (SDS) with different timelines of stress (10 vs 30 d). The authors also compared the DEGs from SDS to RNA-Seq data from cortex of MDD & PTSD postmortem samples. The authors found very limited overlap both within individual datasets and between 10 and 30 d SDS. The authors attribute some of this to variations in the experimental differences (tissue collection, timeline after stress, etc.). The authors find virtually no overlap between the animal models and clinial MDD/PTSD postmortem samples. The authors point to potential differences that may explain the low degree of shared DEG, however the impact of these findings seem to be low. This paper would benefit from a higher amount of analysis of the findings, especially within datasets, that may explain the very low amount of overlap. There are concerns about the clinical postmortem samples that may have been used for this meta-analysis. Finally, because supplementary figures/tables were not available to this reviewer, this manuscript cannot be accurately peer-reviewed.

Dear Referee,

We are thankful to you for reviewing the manuscript and sharing your valuable comments and concerns with us.

Reply: We are very sorry that supplementary materials were not available to you. MDPI journals do not allow these materials to be attached during the manuscript submission process and instead ask authors to upload supplementary materials to special depositories. A link to this depository was enclosed during the submission of the manuscript to this journal.  The Supplementary Materials were uploaded to https://zenodo.org/record/6658135#.Ysq8pYRBxPY

Major criticisms

The authors analyze RNA-Seq datasets from studies with widely ranging differences in experimental procedures (stress procedures, time of tissue collection after SDS, tissue collection methods, etc), which they do point out, however it would benefit to address the degree of overlap between datasets. In other words did multiple studies share the same DEGs, but these DEGs were excluded because one dataset did not? For example, within the 10 d SDS, did 4 of 5 studies have high overlap of DEGs, but the 5th did not therefore these DEGs were excluded. It is unclear in this met-analysis the degree of overlap between each dataset. Some analysis should be made to determine the relationship within a given set of datasets, e.g. 10 d SDS.

Reply: Thanks for your comment. Yes, indeed, the extent of datasets’ overlap is important for preventing outlier results from affecting the overall picture. Supplementary tables 2a–g present lists of differentially expressed genes for each dataset after data reanalysis by our protocol. As the most reproducible genes, one could use those genes that change unidirectionally in most of the experimental datasets (the overlap of the sets of differentially expressed genes). However, we used a meta-analysis algorithm that allows us to more accurately account for the contribution of each study. The presence of differentially expressed genes (padj < 0.05) in the meta-analysis indicates that there are few outliers. The heterogeneity of experimental designs of the original studies will make it difficult to correctly identify the outlying results. Therefore, according to your suggestion, we additionally modified Supplementary tables 3a-b in which we present significant fold changes from the original studies for all differentially expressed genes from the meta-analysis. Our results indicate that most of the differentially expressed genes obtained via the meta-analysis changed expression unidirectionally in at least two original datasets.  

It is unclear why the authors chose only 3 clinical MDD datasets and only 1 clinical PTSD dataset. The authors describe a rigorous method (although see point below) to narrow down SDS datasets and how to integrate them. However, they make very little attempt to do the same with the clinical data. In addition, the MDD and PTSD gene expression data show almost no overlap, which is not consistent with the literature. The authors explain some of the limitations of the preclinical data but fail to describe the limitations of the postmortem data.

Reply: This review was focused on finding molecular mechanisms underlying social defeat stress of various durations and on comparing them with data on post-mortem tissue samples from patients with major depressive disorder or PTSD. Indeed, we did not aim to describe the molecular mechanisms linked with major depressive disorder or PTSD. For this reason, a systematic search for articles and their reanalysis were not performed here. For major depressive disorder, we selected the most recent meta-analysis on post-mortem samples from patients (three datasets included) but no meta-analysis on post-mortem PTSD samples was found. Therefore, we used data from the most relevant original study published in the authoritative journal Nature Neuroscience. As per your comment, in the Discussion section, we have expanded the description of possible limitations of the use of post-mortem data.

 Minor criticisms

The authors used a rigorous method to exclude findings, however some of the reasons for exclusion were not clear: ‘Records removed for other reasons’ during Identification and ‘Records Excluded’ during Screening. This should be explained.

Reply: Yes, you are right, these notes may be confusing. However, these are standard terms according to the PRISMA guidelines. At the first stage, duplicate articles and articles that do not have full-text versions in English, conference abstracts, reviews, or other types of articles (opinion, perspectives, commentaries, etc.) are eliminated. At the second stage, studies that are not relevant in terms of contents are excluded. After compiling a shortlist of articles, we described in detail the reasons why some of them were not included in the analysis (no raw data, and the data were not provided after a request to the corresponding authors; atypical duration of testing of social defeat stress).

Some minor grammatical errors:

  • Incorrect usage of hyphens throughout
  • I think besides is improperly used in this manuscript
  • Table 1 rows are not aligned

Reply: The errors have been corrected.

Reviewer 2 Report

The submitted manuscript “Transcriptome alterations by social defeat stress of various durations in mice and its relevance to depression and post-traumatic stress disorder in humans: a meta-analysis” by Dr V. Reshetnikov and colleagues describes a comparison of transcriptional signatures in the PFC of social defeat stress (SDS) for 10 or 30 days in C57Bl/6 mice to those of the clinical human population with major depressive disorder (MDD) or post-traumatic stress disorder (PTSD). They found that longer exposure to SDS engendered greater transcriptional alterations, and that SDS-susceptible mice show disruption of myelin-related transcripts. Further, the authors show surprisingly little overlap between transcriptomic signatures from postmortem human PFC tissue from individuals with MDD or PTSD to the murine transcriptomic signatures following SDS. The studies described are well-designed, and the findings uncovered contribute to the field as a whole by directly comparing transcriptomic alterations following exposure to SDS in with the clinical population suffering from conditions thought to be modeled by SDS exposure. However, there are a few key points of concern that would improve this manuscript prior to its acceptance for publication.

Major concerns

- My primary concern is the inclusion of two murine studies that are notably different than the other reports included. Specifically:

1. The authors report studies on “C57BL/6 mice,” but do not note substrains. This is concerning, given that substrains are demonstrably different and have varying physiologies (for brief overview, see: https://www.jax.org/news-and-insights/jax-blog/2016/june/there-is-no-such-thing-as-a-b6-mouse). In particular, the Laine et al 2018 study uses the C57BL/6NCrl line, whereas all other reports use the C57BL/6J line. As these two lines are distinct, they should not be directly compared. If this study is to be included, differences between substrains should be mentioned within the discussion.

2. With the exception of one study (Deonaraine et al 2020), all mouse studies were carried out in male animals. Further, in that study alone, the aggressor CD1 mouse was of the opposite sex. While that study did mention removal of aggressor mice that attempted to mate with the female C57 mice, SDS exposure in this study is arguably different than male-male resident/intruder SDS. Because this study is notably different than other studies reported within the meta-analysis, and the only study using females, it at least should be mentioned within the discussion, if not removed from analysis completely.

The authors are encouraged to broaden the discussion of differences between murine studies to address sex and strain. In particular, biological sex should be discussed with regard to comparison to human data.

Minor concerns

- The authors state that, for mouse studies, one of the criteria was that “tissue samples of the cortex and/or PFC were examined.” Table 1 should include what regions (i.e. broad PFC vs subregion vs cortex dissection) were harvested, if specified, within each study. If any studies did not specify beyond “cortex,” they should be excluded.

- The mouse age is listed as “0-12 weeks” for Bondar et al 2018 and Bondar et al 2022 (in press) in Table 1. However, Bondar et al 2018 states “adult male” within the manuscript. Please correct this in the table to indicate that adult animals were used (e.g. 8-12wks).

- Indicate sex of mice for all studies in Table 1, as well as the sex of the aggressor mice.

- It may be helpful to include a table describing human subjects (age, sex, etc) within the 4 studies compared to murine data, or to add this information to Figure 3C.

- Please indicate the inclusion of the authors’ in press study (Bondar et al. 2022) in Figure 1.

- Please add lines between studies in Table 1 to make it easier for readers to differentiate which Data IDs are associated with each study.

- Please note in line 84 of the introduction that human tissue for the PTSD transcriptomic study was obtained from the dorsolateral PFC.

Very minor concerns (typographical errors)

- This is likely due to formatting issues, but a number of words have been unnecessarily hyphenated throughout the manuscript (Ex: “There-fore” line 29; “mo-lecular” line 37; “evalu-ate” line 70; “be-tween” line 232, etc.).

- The use of the word “besides” in lines 110 and 145 is grammatically incorrect.

- Line 310 should say, “…was observed after…” rather than “…was o-served after…”

Author Response

Reviewer 2

The submitted manuscript “Transcriptome alterations by social defeat stress of various durations in mice and its relevance to depression and post-traumatic stress disorder in humans: a meta-analysis” by Dr V. Reshetnikov and colleagues describes a comparison of transcriptional signatures in the PFC of social defeat stress (SDS) for 10 or 30 days in C57Bl/6 mice to those of the clinical human population with major depressive disorder (MDD) or post-traumatic stress disorder (PTSD). They found that longer exposure to SDS engendered greater transcriptional alterations, and that SDS-susceptible mice show disruption of myelin-related transcripts. Further, the authors show surprisingly little overlap between transcriptomic signatures from postmortem human PFC tissue from individuals with MDD or PTSD to the murine transcriptomic signatures following SDS. The studies described are well-designed, and the findings uncovered contribute to the field as a whole by directly comparing transcriptomic alterations following exposure to SDS in with the clinical population suffering from conditions thought to be modeled by SDS exposure. However, there are a few key points of concern that would improve this manuscript prior to its acceptance for publication.

Dear Referee,

We are grateful for your perusal of the manuscript and for providing constructive suggestions.

Major concerns

- My primary concern is the inclusion of two murine studies that are notably different than the other reports included. Specifically:

  1. The authors report studies on “C57BL/6 mice,” but do not note substrains. This is concerning, given that substrains are demonstrably different and have varying physiologies (for brief overview, see: https://www.jax.org/news-and-insights/jax-blog/2016/june/there-is-no-such-thing-as-a-b6-mouse). In particular, the Laine et al 2018 study uses the C57BL/6NCrl line, whereas all other reports use the C57BL/6J line. As these two lines are distinct, they should not be directly compared. If this study is to be included, differences between substrains should be mentioned within the discussion.

Reply: Yes, this is correct, substrains can contribute to the heterogeneity we are seeing. We have added substrain data into Table 1 and now mention in the Discussion section that this factor may have affected the results.

  1. With the exception of one study (Deonaraine et al 2020), all mouse studies were carried out in male animals. Further, in that study alone, the aggressor CD1 mouse was of the opposite sex. While that study did mention removal of aggressor mice that attempted to mate with the female C57 mice, SDS exposure in this study is arguably different than male-male resident/intruder SDS. Because this study is notably different than other studies reported within the meta-analysis, and the only study using females, it at least should be mentioned within the discussion, if not removed from analysis completely.

The authors are encouraged to broaden the discussion of differences between murine studies to address sex and strain. In particular, biological sex should be discussed with regard to comparison to human data.

Reply: We appreciate your suggestion. In the ref. (Deonaraine et al., 2020), we reanalyzed the data related to males only (see the list of samples included in the analysis in Table 1). We have inserted the clarifications into the table.

Minor concerns

- The authors state that, for mouse studies, one of the criteria was that “tissue samples of the cortex and/or PFC were examined.” Table 1 should include what regions (i.e. broad PFC vs subregion vs cortex dissection) were harvested, if specified, within each study. If any studies did not specify beyond “cortex,” they should be excluded.

Reply: The requested clarifications have been added in Table 1.

- The mouse age is listed as “0-12 weeks” for Bondar et al 2018 and Bondar et al 2022 (in press) in Table 1. However, Bondar et al 2018 states “adult male” within the manuscript. Please correct this in the table to indicate that adult animals were used (e.g. 8-12wks).

Reply: Thank you for noticing. This text has been revised.

- Indicate sex of mice for all studies in Table 1, as well as the sex of the aggressor mice.

- It may be helpful to include a table describing human subjects (age, sex, etc) within the 4 studies compared to murine data, or to add this information to Figure 3C.

- Please indicate the inclusion of the authors’ in press study (Bondar et al. 2022) in Figure 1.

- Please add lines between studies in Table 1 to make it easier for readers to differentiate which Data IDs are associated with each study.

- Please note in line 84 of the introduction that human tissue for the PTSD transcriptomic study was obtained from the dorsolateral PFC.

Reply: This information is now mentioned.

Very minor concerns (typographical errors)

- This is likely due to formatting issues, but a number of words have been unnecessarily hyphenated throughout the manuscript (Ex: “There-fore” line 29; “mo-lecular” line 37; “evalu-ate” line 70; “be-tween” line 232, etc.).

- The use of the word “besides” in lines 110 and 145 is grammatically incorrect.

- Line 310 should say, “…was observed after…” rather than “…was o-served after…”

Reply: We have fixed these problems and made grammatical corrections throughout the text.

Round 2

Reviewer 1 Report

The authors failed to make a reasonable effort to address the major criticisms of this reviewer. The authors addressed the major criticisms with very minor changes: they only included the list of p-values from the original data sets in Supplemental Tables and included some discussion of the limitations of the clinical data, however this cannot be reviewed as only half of the words are in English.

The authors report that they are within PRISMA guidelines in reporting exclusionary criteria, however some effort should be made for transparency. In addition, the authors did not make reasonable efforts to address the following PRISMA guidelines that were, to some extent, raised by the reviewer, including:

·      Specify the methods used to decide whether a study met the inclusion criteria of the review, including how many reviewers screened each record and each report retrieved, whether they worked independently, and if applicable, details of automation tools used in the process.

·      Describe the processes used to decide which studies were eligible for each synthesis (e.g. tabulating the study intervention characteristics and comparing against the planned groups for each synthesis

·      Describe any methods used to synthesise results and provide a rationale for the choice(s). If meta-analysis was performed, describe the model(s), method(s) to identify the presence and extent of statistical heterogeneity, and software package(s) used.  

·      Describe any methods used to explore possible causes of heterogeneity among study results (e.g. subgroup analysis, meta-regression).           

·      Describe any sensitivity analyses conducted to assess robustness of the synthesised results.     

·      Describe any methods used to assess certainty (or confidence) in the body of evidence for an outcome.

Author Response

Dear Referee,

We are thankful to you for reviewing manuscript and sharing your valuable comments and concerns with us.

We are very sorry that part of the text was untranslated in Revision 1 version. The text have been corrected.

According to your remarks, the Figure 1 was redrawn and detail description of the exclusion criteria have been added. Records were manually screened independently by two investigators (NB and VR). In addition, to demonstrate homogeneity of studies that included in meta-analysis the Figure 2 with data of principal component analysis (PCA) was added.